# Epidemiology and Diagnosis of Mucormycosis: An Update

**DOI:** 10.3390/jof6040265

**Published:** 2020-11-02

**Authors:** Anna Skiada, Ioannis Pavleas, Maria Drogari-Apiranthitou

**Affiliations:** 1First Department of Medicine, Laiko Hospital, National and Kapodistrian University of Athens, 11527 Athens, Greece; 2Intensive Care Unit, Laiko Hospital, 11527 Athens, Greece; ipavleas@yahoo.com; 3Fourth Department of Internal Medicine, General University Hospital “Attikon”, National and Kapodistrian University of Athens, 12462 Athens, Greece; mdrogari@hotmail.com

**Keywords:** mucormycosis, epidemiology, diagnosis, zygomycosis

## Abstract

Mucormycosis is an angioinvasive fungal infection, due to fungi of the order Mucorales. Its incidence cannot be measured exactly, since there are few population-based studies, but multiple studies have shown that it is increasing. The prevalence of mucormycosis in India is about 80 times the prevalence in developed countries, being approximately 0.14 cases per 1000 population. Diabetes mellitus is the main underlying disease globally, especially in low and middle-income countries. In developed countries the most common underlying diseases are hematological malignancies and transplantation. Τhe epidemiology of mucormycosis is evolving as new immunomodulating agents are used in the treatment of cancer and autoimmune diseases, and as the modern diagnostic tools lead to the identification of previously uncommon genera/species such as *Apophysomyces* or *Saksenaea* complex. In addition, new risk factors are reported from Asia, including post-pulmonary tuberculosis and chronic kidney disease. New emerging species include *Rhizopus homothallicus*, *Thamnostylum lucknowense*, *Mucor irregularis* and *Saksenaea erythrospora*. Diagnosis of mucormycosis remains challenging. Clinical approach to diagnosis has a low sensitivity and specificity, it helps however in raising suspicion and prompting the initiation of laboratory testing. Histopathology, direct examination and culture remain essential tools, although the molecular methods are improving. The internal transcribed spacer (ITS) region is the most widely sequenced DNA region for fungi and it is recommended as a first-line method for species identification of Mucorales. New molecular platforms are being investigated and new fungal genetic targets are being explored. Molecular-based methods have gained acceptance for confirmation of the infection when applied on tissues. Methods on the detection of Mucorales DNA in blood have shown promising results for earlier and rapid diagnosis and could be used as screening tests in high-risk patients, but have to be validated in clinical studies. More, much needed, rapid methods that do not require invasive procedures, such as serology-based point-of-care, or metabolomics-based breath tests, are being developed and hopefully will be evaluated in the near future.

## 1. Introduction

Mucormycosis is an angioinvasive fungal infection due to fungi of the order Mucorales. Depending on the clinical presentation it is classified as rhinocerebral, pulmonary, cutaneous, gastrointestinal, disseminated or other, which includes uncommon rare forms, such as endocarditis, osteomyelitis, peritonitis, renal, etc. The disease was first described in 1876 when Fürbinger described in Germany a patient who died of cancer and in whom the right lung showed a hemorrhagic infarct with fungal hyphae and a few sporangia [1]. In 1885, Arnold Paltauf published the first case of disseminated mucormycosis, which he named “Mycosis mucorina” [2]. His drawings of the etiologic agent showed the presence of sporangiophores and rhizoid-like structures, and this led to the conclusion that the infection was most probably caused by *Lichtheimia corymbifera*. Over time, more cases were diagnosed, and the incidence of the disease has increased [3]. Currently, Mucorales fungi are the next most common mold pathogens after *Aspergillus*, leading to invasive fungal disease in patients with malignancies or transplantation [4]. The incidence of mucormycosis has also increased significantly in patients with diabetes [5], which is the commonest underlying risk factor globally. Τhe epidemiology of mucormycosis is evolving as new immunomodulating agents are used in the treatment of cancer and autoimmune diseases, and as the modern diagnostic tools lead to the identification of previously uncommon genera/species such as the *Apophysomyces* [6] or *Saksenaea* complex [7]. The aim of this article is to present an update on the epidemiology and the available diagnostic methods for this potentially lethal disease.

## 2. Epidemiology

Most of the data regarding the epidemiology of mucormycosis originate from case reports and case series. The first extensive review of the literature was made by Roden et al. in 2005 [8]. The analysis comprised 929 cases published from 1940 to 2003, providing useful information about the disease, but it also included cases of entomophthoramycosis. More recently, Jeong et al. analyzed 851 cases published from 2000 to 2017. In this study, the review was undertaken and reported using the PRISMA (Preferred Reporting Items for Systematic reviews and Meta-Analyses) guidelines and it included only cases caused by Mucorales [9]. Other relatively large case series were collected either on a national level [10,11,12] or in patients with selected underlying diseases, for example, hematopoietic stem cell transplantation [13]. Registries are another source of valuable information, despite their inherent limitations. The Working Group on Zygomycosis of the European Confederation of Medical Mycology (ECMM) and the International Society of Human and Animal Mycology (ISHAM) constructed such a registry in 2004 (www.zygomyco.net) and published 230 cases from Europe in 2011 [14].

### 2.1. Incidence of Mucormycosis

The incidence of mucormycosis is increasing [3,5,15,16,17]. The exact incidence/prevalence is not known because there are few population-based studies [3,15,18], and they differ in capture periods, populations and definition or diagnostic procedures. In the study performed in the San Francisco Bay area, from 1992 to 1993, the annual incidence of mucormycosis was found to be 1.7 cases per 1 million individuals (500 cases per year) [15]. In the study published by Kontoyiannis et al. in 2016, “the prevalence of mucormycosis-related hospitalizations was estimated as 0.12 per 10,000 discharges during January 2005–June 2014, and it increased to 0.16 per 10,000 discharges if the definition of mucormycosis was relaxed to not require the use of amphotericin B or posaconazole” [18]. In a national population-based study in France, the prevalence of mucormycosis was 0.7 cases per million in 1997 and rose to 1.2 per million in 2006 [10]. From Belgium, Saegeman et al. reported a rise from 0.019 cases/10,000 patient-days during 2000 to 0.148 cases/10,000 patient-days during 2009 [16]. In India, the prevalence of mucormycosis is approximately 0.14 cases per 1000 population, which is about 80 times the prevalence of mucormycosis in developed countries [7]. All the above publications highlight the fact that mucormycosis is an emerging disease. Roden et al. included a figure where the rise of incidence over 6 decades (1940–1999) is clearly shown (8). The same authors reported the rise of percentages of cases documented by culture and by decade. The percentage was 50% in the 1980s, and 71% in the 2000s. This may indicate that, in addition to an actual increase of the incidence of cases, better diagnostic methods or increased awareness of the disease play an important role. The rising trend of mucormycosis was also shown by Chakrabarti et al. who published three consecutive studies from a single center in India; the incidence of mucormycosis was 12.9 cases/year over the first decade [19], 35.6 cases/year over a 5-year period [5] and, during an eighteen months period, 50 cases/year [20].

### 2.2. Causative Agents

The order Mucorales comprises 261 species in 55 genera, 38 of which have been associated with human infections [21]. Their taxonomy has changed widely the last years due to molecular phylogenetic studies and, inevitably, some taxa have undergone several name changes as well. The current nomenclature of these taxa is presented in Table 1.

The fungal spores enter the human organism by inhalation, ingestion or direct inoculation. The most common species all over the world is *Rhizopus arrhizus* (formerly *Rhizopus oryzae*). Other isolated fungi belong to the genera *Lichtheimia, Mucor, Rhizomucor, Cunninghamella, Saksenaea*, *Apophysomyces, Cokeromyces, Actinomucor* and *Syncephalastrum*. In the global review by Jeong et al. *Rhizopus* spp., *Lichtheimia* spp. and *Mucor* spp. accounted for 75% of all cases [9]. The agents of mucormycosis vary depending on the geographical area. In the ECMM study from Europe, *Rhizopus* spp. were isolated in 34% of cases, *Lichtheimia* spp. in 19% and *Mucor* spp. in 19% [14]. In the RetroZygo study from France, *Rhizopus* spp. were the causative agents in 52%, while the second most common genus was *Lichtheimia* (29%) [10]. In the studies from India, the second most commonly isolated agents were *Apophysomyces* spp. [5,22]. *Apophysomyces* and *Saksenaea* spp. are almost always responsible for cutaneous mucormycosis in immunocompetent patients, and are mainly found in Asia [6,22]. New species are emerging, including *Rhizopus homothallicus* [23], *Thamnostylum lucknowense* [24], *Mucor irregularis* [25] and *Saksenaea erythrospora* [7].

### 2.3. Predisposing Factors/Underlying Conditions

The most important conditions that predispose to mucormycosis, according to various studies (Table 2), include diabetes mellitus (DM), with or without ketoacidosis, hematological malignancies (HM), other malignancies, transplantation, prolonged neutropenia, corticosteroids, trauma, iron overload, illicit intravenous drug use, neonatal prematurity and malnourishment [26]. Immunocompetent patients can also be affected, when the spores of the fungus are directly inoculated in the skin, as a result of trauma or burns.

Risk factors for mucormycosis vary considerably by geographical area. In studies from Europe [10,14,27] the most common underlying disease was a hematological malignancy, while in India [5,11,20,29], Iran [29,31] and Mexico [12] it was diabetes mellitus. Diabetes was also the leading underlying condition in the study by Stemler et al. in countries of the Middle East and North Africa [34].

Several studies have shown that the underlying disease is correlated to the site of infection [8,9,14]. Hematological malignancies and neutropenia are associated with pulmonary mucormycosis and diabetes mellitus with sinusitis and rhinocerebral disease, while trauma usually leads to cutaneous mucormycosis.

#### 2.3.1. Diabetes Mellitus and Ketoacidosis

Diabetes mellitus is the leading underlying disease in patients with mucormycosis globally [8,9]. According to the World Health Organization (WHO) “the global prevalence (age-standardized) of diabetes has nearly doubled since 1980, rising from 4.7% to 8.5% in the adult population. Globally, an estimated 422 million adults were living with diabetes in 2014, compared to 108 million in 1980” [35]. Diabetes prevalence has risen faster in low- and middle-income countries than in high-income countries. The number of people aged 20–79 years with diabetes in 2011 was 61.3 million in India, and it is estimated to rise to 101.2 million in 2030 [36]. A great rise in the diabetic population is also predicted for China, Brazil, Japan, Mexico, Egypt and Indonesia [37]. Accordingly, the cases of mucormycosis are expected to increase. In the latest review by Jeong et al. diabetes mellitus was the most common underlying condition in 40% of cases and 20% had documented ketoacidosis [9]. Uncontrolled, type II, diabetes is the most common type in diabetic patients with mucormycosis. In a recent study comparing North and South India, diabetic ketoacidosis was found in 90% of cases from North India and 10% of cases from South India [29]. Diabetes has been reported as a risk factor for mucormycosis in 73.5% of cases in India [11], 75% [30] in Iran and 72% [12] in Mexico. In contrast, the percentages from the European ECMM study were 17% [14], from Italy 18% [27], from France 23% [10] and from Lebanon 35% [31]. In the Indian publications, mucormycosis was the unmasking disease for diabetes mellitus in 12–31% of patients [5,11,22,38].

#### 2.3.2. Hematological Malignancy and Hematopoietic Stem Cell Transplantation

Hematological malignancies (HMs) and hematopoietic stem cell transplantation (HSCT) are the most common underlying diseases in mucormycosis in Europe, USA and Australia (Table 1). In India HM was a risk factor in 1–9% [5,11,22,29] and in Iran in 3% [30]. The majority of patients with HM and mucormycosis in various studies had acute myeloid leukemia (48% in the ECMM study [14], 46% in Italy [27], 34% in France [10], 38% in Lebanon [31] and 42% in the global review by Jeong et al. [9]). The rest of the patients with HM had acute lymphoblastic leukemia, non-Hodgkin’s lymphoma, myelodysplastic syndrome and other, more rare, malignancies. The risk is higher when the patient has prolonged neutropenia [39]. HSCT is also an important risk factor. In the global analysis of mucormycosis in France from 2005 to 2007, HSCT was the underlying condition in 12% of cases [10]. Conversely, in a French nationwide retrospective study from 2003 to 2008, the prevalence of mucormycosis among HSCT recipients was found to be 0.4% [40]. In order to investigate the epidemiology of infections due to Mucorales and other molds in transplant recipients, Park et al. analyzed the data from the Transplant-Associated Infection Surveillance Network (TRANSNET), where 23 transplant centers in USA conducted a prospective surveillance for invasive fungal infections during 2001–2006 [41]. The 12-month mucormycosis cumulative incidence was 0.29% for HSCT. Mucormycosis incidence among HSCT recipients varied widely, from 0.08% to 0.69% [41]. The incidence of HSCT as a risk factor for mucormycosis in developing countries was reported as 1% for India [11], 2% for Iran [30] and 2% for South America countries [28].

#### 2.3.3. Solid Organ Malignancies and Solid Organ Transplantation

Solid organ malignancies (SOMs) and solid organ transplantation (SOT), although not as common as HM and HSCT, are also important risk factors for mucormycosis. The reported prevalence varies, ranging from 0.6% to 23% (Table 1), but the studies cannot be compared because the methods of collecting data are different. The numbers are higher in reviews of the literature, where there is publication bias [8,9,28,33]. In the RetroZygo study from France SOT was the underlying disease in 3% of cases [10]. In the TRANSNET analysis the 12-month mucormycosis cumulative incidence for SOT was 0.07% and mucormycosis accounted for 2% of all invasive fungal infections in these patients [41,42]. The results were similar in the Prospective Antifungal Therapy Alliance (PATH Alliance) study, where 7526 invasive fungal infections in a total of 6918 adults SOT recipients were included [43]. Mucormycosis accounted for 1.6% of these infections. The incidence also depends on the organ being transplanted. In the review by Almyroudis et al. the incidence in renal transplant recipients was 0.4–0.5, in liver recipients 4–16, in heart recipients 8 and in lung recipients 13.7–14, all per 1000 patients [44]. In a prospective, matched case-control study, renal failure, diabetes mellitus and prior voriconazole and/or caspofungin use were associated with a higher risk of mucormycosis, whereas tacrolimus, a calcineurin-inhibitor, was associated with a lower risk [45]. Calcineurin plays a vital role in the virulence and pathogenicity of several opportunistic fungi [46] and its involvement in the virulence of Mucorales is currently a hot topic of research [47].

#### 2.3.4. Corticosteroids and Other Immunosuppressive Agents

Chronic administration of corticosteroids and other immunosuppressive agents is an important risk factor for mucormycosis. They are used in the treatment of malignancies, transplantation and autoimmune diseases. Corticosteroids impair migration, ingestion and phagolysosome fusion in macrophages. In addition they may lead to drug-induced diabetes. Prolonged (>3 weeks) high-dose systemic corticosteroids are risk factors for mucormycosis [48]. However, there have been reports of mucormycosis associated with short courses of corticosteroids [49]. In the ECMM European study 46% of patients had received corticosteroids in the month prior to diagnosis of mucormycosis and 44% had received other immunosuppressive agents [14]. In the global review by Jeong et al. [9] 3% of patients had an autoimmune disease, while in Australia, Kennedy et al. [33] reported that 12% had an autoimmune disease. There are few cases of systemic lupus erythematosus associated mucormycosis in the literature, but the mortality of these infections is exceedingly high (80%) [50]. There have also been reports of mucormycosis in other autoimmune diseases, where the infection mimicked the relapse of Wegener’s granulomatosis [51] or, in another case, was confused with giant cell arteritis [52]. Although these cases are extremely rare, they may be underdiagnosed and mucormycosis should be included in the differential diagnosis.

#### 2.3.5. Iron Overload

Increased serum iron is a risk factor for mucormycosis, as iron plays a crucial role in the pathogenesis of this infection [53]. Iron is normally attached to transferrin and ferritin, and is not available to the Mucorales fungi. In patients with diabetic ketoacidosis or other forms of acidosis there is decreased affinity of these proteins to bind iron [54]. Serum iron may also be increased in patients undergoing dialysis or multiple transfusions. In the past, the iron chelator deferoxamine was used in these cases. This is a bacterial siderophore and is actually utilized by Mucorales as a xenosiderophore for acquiring iron from the host. It soon became apparent that deferoxamine was a risk factor for mucormycosis, most often disseminated [55]. The newer chelators, deferasirox and deferiprone, do not have xenosiderophore activity. Iron overload has also been implicated in the increased susceptibility of liver transplant recipients to disseminated mucormycosis [56].

#### 2.3.6. Breakthrough Mucormycosis

Another factor that may predispose to mucormycosis is the use of antifungal prophylaxis or treatment, which is effective against *Aspergillus* but not Mucorales (voriconazole and echinocandins) [57,58,59,60]. The first reports of possible association of prophylaxis with voriconazole and mucormycosis were published in 2004 [58,59,60]. Marty et al. hypothesized that a possible explanation for this was prolonged use of voriconazole or increased survival among profoundly immunosuppressed patients, i.e., patients who had received HSCT [58]. Studies in animal models suggested that voriconazole might increase the virulence of certain Mucorales [61]. Breakthrough invasive fungal infections continued to emerge when the newer azoles, posaconazole and isavuconazole were introduced, despite their activity against Mucorales [62,63,64]. In a single-center study from Austria 13% of HSCT patients receiving prophylaxis with posaconazole developed breakthrough invasive fungal infections and 55% of them were due to mucormycetes [62]. A recent review by Lionakis et al. showed that the incidence and spectrum of breakthrough invasive mold infections vary significantly depending on the specific mold-active antifungal used for prophylaxis, local epidemiology and patient characteristics [65].

#### 2.3.7. Other

Other diseases associated with mucormycosis are intravenous drug use, AIDS, renal failure, liver diseases, chronic alcoholism, malnutrition and low birth weight infants [9]. Mucormycosis in patients who are HIV positive is extremely rare. In an old retrospective study of 1630 autopsies of patients who died of AIDS from 1984 to 2002, only 2 patients had mucormycosis [66]. In the recent review of the literature by Moreira et al. only 4 out of 67 cases (5.9%) had no other risk factors [67]. The commonest comorbidities were a history of intravenous drug use (IVDU; 50%), neutropenia (29.7%) and corticosteroid use (25%).

Patients with a history of IVDU who develop mucormycosis, most often present with isolated cerebral infection [8]. Conversely, in a review of 68 patients with isolated cerebral mucormycosis, 82% had a history of IVDU, and the authors concluded that the presence of lesions in the basal ganglia, rapidly progressive symptoms and a history of IVDU should raise suspicion for mucormycosis and for initiation of amphotericin B and stereotactic aspiration [68]. In a study from India, published in 2019, post-pulmonary tuberculosis (6.9%) and chronic kidney disease (8.9%) were emerging risk factors [29].

#### 2.3.8. No Underlying Disease

In a significant proportion of cases mucormycosis develops in immunocompetent patients. In the two large reviews by Roden et al. and Jeong et al., 19% and 18% of patients respectively, had no underlying disease. In such cases trauma or burns is the usual predisposing factor, resulting in cutaneous disease. The trauma can be minor (injection sites, animal bites, gardening, etc.) or major, including motor vehicle accident, natural disasters and surgery [69]. In cases associated with natural disasters uncommon species have been isolated: wound infections due to *Apophysomyces elegans* were reported after the tsunami in Sri-Lanka [70] and *Syncephalastrum racemosum* was isolated from respiratory samples following hurricane Katrina [71].

Mucormycosis has also been reported following combat-related injury [72]. Another clinical entity, reported mainly from India and China, is isolated renal mucormycosis in immunocompetent patients [73]. Finally, there have been several reports from Asia, about an infection in immunocompetent patients by *Mucor irregularis* (formerly *Rhizomucor variabilis*), leading to primary cutaneous mucormycosis, which progresses very slowly, in contrast to the classical presentation of mucormycosis [25,74].

#### 2.3.9. Healthcare Associated

There have been multiple reports of healthcare-associated mucormycosis, either as isolated cases or as outbreaks. In a publication from India, 75 cases of mucormycosis were reported during an eighteen-month period, of which 9% were nosocomial [20]. Healthcare-associated mucormycosis has been attributed to various exposures in the hospital environment [75]:(a)The use of non-sterile products is the most commonly suspected cause of infection [76]. Bandages, adhesives, nitroglycerin patches [77], contaminated linen [78], wooden tongue depressors, ostomy bags [79] and probiotics have all been implicated. There has even been a report of an outbreak due to allopurinol tablets and prepackaged food [80].(b)Various procedures and medical devices, such as catheters, insulin pumps and finger sticks [81], and insertion of tubes, tooth extractions [82] and surgery.(c)Environmental factors may also be a source of infection. Molds may be found in the air, dust, water or any surfaces in the hospital. Construction works increase the risk of invasive fungal infections. Outbreaks have been linked to defective ventilation systems and water leakage.

In the analysis of 169 cases by Rammaert et al. 22% of the patients had diabetes mellitus, 24% had a solid organ transplant, 12% had a malignancy, 37% had a history of prolonged steroid therapy and 5% had other immunodepression factors [75].

The clinical presentation varies, depending on the source of infection. Infections due to bandages, adhesives or contaminated wound dressings are mostly cutaneous. Percutaneous exposure in immunocompromised patients has led to disseminated disease. Inhalation leads to pulmonary and rhino-cerebral infection, while ingestion of tablets or food, and the use of tongue depressors, are responsible for gastrointestinal mucormycosis. Dialysis catheters have been linked to peritonitis [75].

## 3. Diagnosis

Early diagnosis of mucormycosis is of utmost importance, since it may improve outcome. Studies have shown that it increases survival [83], and it may also reduce the need for or extent of surgical resection, disfigurement and suffering [84]. Since the disease is rare, a high index of suspicion is very important. Diagnosis consists of recognition of risk factors, assessment of clinical manifestations, early use of imaging modalities and prompt initiation of diagnostic methods based on histopathology, cultures and advanced molecular techniques.

### 3.1. Clinical Diagnosis

The clinical approach to diagnosis has low sensitivity and specificity. The hallmark of mucormycosis is tissue necrosis resulting from angioinvasion and thrombosis, the absence, however, of a necrotic eschar does not preclude the diagnosis. Necrotic cutaneous lesions in immunocompromised patients may be due to mucormycosis, but the differential diagnosis includes other pathogens, such as *Aspergillus*, *Fusarium*, *Pseudallescheria* and *Scedosporium* species [84]. *Pseudomonas aeruginosa* causing ecthyma gangrenosum may also produce a similar picture. Although the clinical signs and symptoms are non-specific, some of them may have a potentially high predictive value. In a susceptible host, a history of prophylaxis with voriconazole or the emergence of breakthrough fungal infection while receiving agents active against *Aspergillus* but not Mucorales, suggests the presence of mucormycosis [58,59,60,85].

A patient with diabetes and sinusitis should be thoroughly examined for possible mucormycosis. Corzo-Leon et al. [12] proposed an algorithm for the diagnosis and treatment of rhino-orbito-cerebral mucormycosis in patients with diabetes mellitus. The “red flags/warning signs” in this algorithm are cranial nerve palsy, diplopia, sinus pain, proptosis, periorbital swelling, orbital apex syndrome or a palatine ulcer. The finding of any of these signs should prompt immediate further testing, including blood tests, imaging, ocular and/ or sinus surgery or endoscopic revision and initiation of antifungal treatment. Although diabetes, with or without ketoacidosis, is the usual underlying disease in rhinocerebral mucormycosis, it should be noted that it might also occur in patients with malignancies, recipients of HSCT or SOT and other immunocompromised individuals [26]. Pulmonary mucormycosis most often occurs in neutropenic patients. The clinical features cannot be easily distinguished from those of pulmonary aspergillosis or fusariosis. Additionally, in countries where tuberculosis is endemic, the two infections may coexist [86]. Prolonged fever, not responding to broad-spectrum antibiotics, is usually present. Nonproductive cough is a common symptom, whereas hemoptysis, pleuritic chest pain and dyspnea are less common. Imaging studies have an important role in the differentiation of invasive fungal infections. Chamilos et al. reported that multiple (≥10) nodules, and pleural effusion are more common in mucormycosis [85]. Another finding that indicates mucormycosis is the reverse halo sign (RHS) on computerized tomography (CT) scan. In the study by Legouge et al. sequential thoracic CT scans were performed in leukemic neutropenic patients, and the RHS was observed in 15 of 16 patients (94%) during the first week of the disease, while other radiologic findings, such as multiple nodules, appeared later. These findings led the authors to the conclusion that in the particular setting of neutropenic leukemic patients with pulmonary infection, the presence of the RHS on CT was a strong indicator of pulmonary mucormycosis [87]. This was also shown in another study, where the CT scans of 24 patients with pulmonary mucormycosis were compared to those of 96 patients with invasive lung aspergillosis. The RHS was observed in 54% of patients with mucormycosis and only in 6% of patients with aspergillosis [88], whereas some airway-invasive features, such as clusters of centrilobular nodules, peribronchial consolidations and bronchial wall thickening, were more common in patients with invasive aspergillosis. In a more recent study, however, the most common pattern on initial CT scans of immunocompromised patients with pulmonary mucormycosis was nodule (≤3 cm)/mass (>3 cm) or consolidation with surrounding ground-glass opacity halo (18/20, 90%) [89]. On follow-up CT, morphologic changes could be seen in 87% and they included RHS, central necrosis and air-crescent sign. Sequential morphologic changes were related to the absolute neutrophil count in 13 of 15 patients [89]. Another imaging technique, which is not yet widely used, is the positron emission tomography-computed tomography (PET/CT) with [18F]-fluorodeoxyglucose (FDG) [90]. While all these studies point towards the diagnosis of mucormycosis, definite diagnosis requires laboratory methods, such as histopathology and cultures.

### 3.2. Routine Laboratory Diagnosis

In clinical practice, laboratory diagnosis of mucormycosis includes histopathology, direct examination of wet mounts and culture.

#### 3.2.1. Histopathology

A definitive diagnosis is based on the demonstration of fungal hyphae typical for mucormycetes in biopsies of affected tissues, or bronchoalveolar lavage (BAL) in patients with pulmonary mucormycosis. Histopathology is a very important diagnostic tool since it distinguishes the presence of the fungus as a pathogen in the specimen from a culture contaminant and is indispensable to define whether there is blood vessel invasion [91]. It can furthermore reveal coinfections with other molds. *Mucorales* genera produce typically non-pigmented, wide (5–20 μm), thin-walled, ribbon-like hyphae with no or few septations (pauciseptate) and right-angle branching [92], in contrast to those of the *Aspergillus* species or other hyaline molds, which are typically 3–5 μm wide, septate and form acute-angle branching. Routine hematoxylin and eosin (H&E) stains may show only the cell wall with no structures inside, or occasionally, very degenerate hyphae. Stains that can help highlight the fungal wall include Grocott methenamine-silver (GMS) and periodic acid-Schiff PAS stains, although PAS gives a better visualization of the surrounding tissue compared to GMS [91].

#### 3.2.2. Direct Microscopy

For a rapid presumptive diagnosis of mucormycosis direct microscopy of KOH wet mounts can be used. It can be applied to all materials sent to the clinical laboratory, preferably using fluorescent brighteners such as Blankophor and Calcofluor White together with KOH, which enhance the visualization of the characteristic fungal hyphae, in this case requiring a fluorescent microscope [84]. Direct microscopy of fresh material is an inexpensive, yet invaluable method to rapidly give a presumptive diagnosis and to define clear surgical margins for invasive fungal infection intraoperatively, [93] and it is strongly recommended, along with histopathology, by a panel of experts of the European Confederation of Medical Mycology in cooperation with the Mycoses Study Group Education and Research Consortium (ECMM/MSG ERC) [94]. These methods, however, are not able to identify a fungus to the genus or species level. Another method, immunohistochemistry using monoclonal antibodies against *R. arrhizus* (recently commercially available) can aid in the diagnosis when cultures are negative and has been proven useful for differentiating aspergillosis from mucormycosis (sensitivity 100%, specificity 100% for mucormycosis) [95,96,97] and has gained a moderate recommendation of B IIu in the recent ECMM/MSG ERC guidelines [94].

#### 3.2.3. Culture

Culture of specimens is essential for the diagnosis of mucormycosis since it allows identification to the genus and species level, and eventually antifungal susceptibility testing. Most medically important *Mucorales* are thermotolerant and are able to grow rapidly at temperatures of 37 °C. They grow on virtually any carbohydrate substrate, colonies appearing usually within 24–48 h and identification is based on colonial and microscopic morphology and growth temperature [84]. Matrix assisted laser desorption ionization-time of flight mass spectrometry (MALDI-TOF) identification of cultured *Mucorales* is a promising method for those laboratories that are accordingly equipped [98], but the commercially available databases should be expanded further, and more validation data are needed. Until then, molecular identification remains the gold standard [94].

A positive culture from a sterile site confirms the diagnosis, while a positive culture from a non-sterile site could be due to a contaminant and must be combined with clinical and radiological data to establish a probable diagnosis [94]. Hence, there is a caveat for falsely positive results, especially when histopathology is not available.

The major concern about culture, however, is its low sensitivity, as it can be falsely negative in up to 50% of mucormycosis cases [84,99]. This can be attributed to a number of reasons, such as grinding or homogenization of tissue specimens, which may destroy the delicate hyphae of mucormycetes, the presence of genera that require special culture conditions, recent or ongoing therapy with antifungals effective on *Mucorales*, [84,99] or even a lack of expertise. Proper sampling and handling of the specimens before examination are a prerequisite for an optimal yield. Therefore, upon suspicion of a case, good communication and close collaboration between clinicians and the microbiology laboratory is essential to ensure that all steps of the diagnostic procedure will be taken properly.

### 3.3. Applied and Emerging Molecular Methods

Molecular methods have evolved as a useful tool to confirm the infection and identify the strains involved. There are methods therefore, developed on the one hand to accurately identify to the species level strains that already have grown in cultures, and on the other hand methods to detect mucormycetes in tissues.

The ITS region is the most widely sequenced DNA region for fungi in general. ITS sequencing is a reliable method and has typically been the most useful for molecular systematics at the species level, and even within species and it is recommended as a first-line method for species identification of *Mucorales* [94].

For the detection in tissues, several methods have been developed, including PCR based techniques such as nested PCR, real-time PCR (qPCR), nested PCR combined with RFLP [100], PCR coupled with electrospray ionization mass spectrometry (PCR/ESI-MS) [101] and PCR/high-resolution melt analysis (HRMA) [102]. Many of these methods have been reportedly successfully applied, and perform better on fresh or deep-frozen samples than on paraffin embedded tissues. Target selection for the PCR is a key determinant of the success of the method [103]. There are methods that use pan-fungal primers targeting the ITS genomic region with the following sequencing of the amplified DNA, or methods of multiplex PCR using specific primers targeting a restricted number of mucoralean genera/species. Most of the molecular tests target the 18S ribosomal RNA genes, but also other targets have been investigated. These include the 28S rDNA [104], the mitochondrial gene *rnl* [105], the cytochrome b gene [106] or the Mucorales-specific CotH gene [107].

Kasai et al. developed a specific qPCR targeting the 28S rDNA [108], Bernal-Martinez et al. a multiplex real-time qPCR targeting the ITS1/ITS2 region with specific probes for *R. oryzae*, *R. microsporus* and *Mucor* spp. [109] and Springer et al. [104] developed two independent *Mucorales* specific real-time qPCR assays, targeting two different regions of the multicopy ribosomal operon-18S and 28S, that are able to detect DNA from a broad range of clinically relevant *Mucorales* species.

The in-house tests that have been developed so far use different probes and primers, therefore the lack of standardization makes their implementation in the clinical laboratory difficult. Furthermore, they are cumbersome, costly and need expertise. A new pan-Mucorales real-time (qRCR) commercial kit (Mucorgenius^®^, PathoNostics, Maastricht, The Netherlands) appeared to be a fast diagnostic test with an overall sensitivity of 75% tested on serial blood samples from patients with culture-positive invasive mucormycosis, often preceding the final diagnosis by several days to weeks [110]. Guegan et al. have validated its performance comparing it with four in-house qPCR assays targeting the 18S rDNA in pulmonary samples, allowing the detection of four main clinically relevant genera, i.e., *Mucor-Rhizopus* spp., *Lichtheimia* spp., *Rhizomucor* spp. and *Cunninghamella* spp. [111]. The commercial assay was user friendly and all qPCR assays used in the study were superior to conventional methods to detect fungal DNA. Drawbacks of the method are that genus identification is not possible, and that it may not detect low fungal burden infections [111].

Clinical studies are required for the validation of all these methods. In samples with the characteristic hyphae of *Mucorales* upon histopathological examination, the application of molecular methods confirms the diagnosis, therefore are highly recommended [94].

Recent studies have confirmed the value of qPCR in BAL, with regard to earlier diagnosis and better outcomes in cases of pulmonary mucormycosis [102,103,104,105,106,107,108,109,110,111,112,113].

### 3.4. Non-Invasive Diagnostic Methods: An Eye to the Future?

#### 3.4.1. Molecular

The samples used in most of the above-mentioned methods presuppose invasive procedures, which may not be applicable to some patient groups (hematologic malignancies with thrombocytopenia, ICU patients, etc.). Despite the angioinvasive nature of the infection, blood cultures remain negative. In fact, only a handful of cases with a positive blood culture have been described so far. [114,115,116,117]. However, fungal DNA circulates in the blood. There is therefore a lot of ongoing research, focusing on non-invasive methods, such as qPCR for the detection of circulating mucoralean DNA in blood (plasma or serum) or urine [104,107,118].

Serum *Mucorales* PCR has been shown to be a highly reliable tool for the diagnosis of invasive mucormycosis in immunocompromised patients. Millon et al. in a study using three qPCR (for *Absidia corymbifera* (*Lichtheimia*), *Mucor/Rhizopus* and *Rhizomucor* 18S ribosomal RNA genes) on sera of mucormycosis patients showed that this method was highly sensitive, had a low detection level and could detect infection 3–68 days earlier than the conventional methods [119]. These findings were confirmed in a following study of this group in collaboration with ten other centers of the French network for surveillance of invasive fungal infection (RESSIF, Institut Pasteur, Paris) [120]. This method could be also useful for treatment monitoring [120]. Legrand et al., using a qPCR method, found that detection of circulating *Mucorales* DNA (cmDNA) allowed earlier diagnosis of invasive wound mucormycosis (IWM) in severely ill burn patients and earlier treatment initiation. [121]. Springer et al. also found early detection (21 days earlier compared to conventional methods) of *Mucorales* in blood and tissue samples from hematological patients with invasive mucormycosis, using a fast (4 h turnaround time) probe-based *Mucorales*-specific real-time PCR assay (Muc18S) [122].

Methods of qPCR in blood are rapid (about 3 h turnaround time). Although their sensitivity is lower than in tissues, they are highly specific. Furthermore, they are able to diagnose mucormycosis in immunocompromised patients earlier than the conventional mycological methods (8 days) or imaging (3 days [119,120,122]. They are appropriate therefore for screening and monitoring of patients at high-risk and could increase survival [123].

In the search of new specific targets, Baldin et al. [107] have shown that the spore coating protein homolog *CotH* genes, which are unique for *Mucorales*, has given encouraging results in a mouse model, with urine being the preferred sample type compared to plasma or BAL. In cases of confirmed mucormycosis this method had 90% sensitivity and was 100% specific.

#### 3.4.2. Serology

There are no commercially available antigen markers to detect *Mucorales*, as galactomannan (GM) for *Aspergillus*. However, GM testing in blood and BAL in hematology patients or patients with compatible chest CT imaging results may be used to decrease the likelihood of mucormycosis [94]. Furthermore, the panfungal β-d-glucan test does not detect antigen components of the *Mucorales*.

In search of new targets for serologic tests, Burnha-Maurish et al. evaluated a monoclonal antibody (2DA6) in a sandwich ELISA and found it to be highly reactive with purified fucomannan of the *Mucor* spp. [124].

They subsequently constructed a lateral flow immunoassay (LFIA) for detection of *Mucorales* cell wall fucomannan in clinical samples and demonstrated that LFIA was more convenient to use than ELISA and had the potential to be used as a point-of-care test on BAL, serum, urine and tissue. The test was able to rapidly and accurately identify *R. delemar*, *L. corymbifera*, *M. circinelloides* and *C. bertholletiae*, early after infection (within 3–4 days of infection) in murine models [125].

#### 3.4.3. Metabolomics-Breath Test

In an experimental murine model of invasive mucormycosis (IM), Koshy et al. examined breath volatile metabolite profiles, using the three *Mucorales* species that most commonly cause human IM—*Rhizopus arrhizus var. arrhizus*, *R. arrhizus* var. *delemar* and *R. microsporus*, by thermal desorption gas chromatography/tandem mass spectrometry (GC–MS) [126]. Mice infected with *Aspergillus fumigatus* were used as controls. They also analyzed breath volatile metabolites from five patients eventually diagnosed with proven IM caused by *R. microsporus*, sampled prospectively. The findings showed that the three *Mucorales* species had distinct breath profiles of the volatile metabolite sesquiterpene, which could be used to identify these infections in vivo. These profiles distinguished the infections from each other and from aspergillosis, therefore this method has the potential to diagnose fungal infection non-invasively, and perhaps monitor response to therapy. Additionally, it could be used in a high risk population such as patients with neutropenia due to treatment for leukemia or those undergoing hematopoietic cell transplantation, to screen for mold infections additionally to *Aspergillus* galactomannan. This method appears to be very appealing and promising, but needs further evaluation.

## 4. Conclusions

Epidemiology of mucormycosis is evolving. In light of new evidence, diabetes mellitus remains the main underlying disease globally. As diabetes is rising, especially in low and middle-income countries, a rise in mucormycosis cases is expected and this should be alarming. In developed countries, the most common underlying diseases are hematological malignancies, with acute myeloid leukemia the most frequent. IVDU is the main risk factor for isolated cerebral mucormycosis, and post-pulmonary tuberculosis and chronic kidney disease are emerging risk factors in developing countries. New emerging species include *Rhizopus homothallicus*, *Thamnostylum lucknowense*, *Mucor irregularis* and *Saksenaea erythrospora*.

Diagnosis of mucormycosis remains challenging. Histopathology, direct examination and culture remain essential tools, although the molecular methods are improving. New molecular platforms are being investigated and new fungal genetic targets are being explored. Molecular-based methods have gained acceptance for confirmation of the infection when applied to tissues. Methods on detection of Mucorales DNA in blood have shown promising results for earlier and rapid diagnosis and could be used as screening tests in high-risk patients, but have to be validated in clinical studies. More, much needed, rapid methods that do not require invasive procedures, such as serology-based point-of-care, or metabolomics-based breath tests, are being developed and hopefully will be evaluated in the near future.

## Figures and Tables

**Table 1 jof-06-00265-t001:** Current nomenclature of medically important mucoralean species according to updated taxonomy [21].

Current Species Names	Previous Names/Synonyms
*Lichtheimia corymbifera*	*Absidia corymbifera, Mycocladus corymbifer*
*Lichtheimia ornata*	*Absidia ornata*
*Lichtheimia ramosa*	*Absidia ramosa, Mycocladus ramosus*
*Mucor ardhlaengiktus*	*Mucor ellipsoideus, Mucor circinelloides f. circinelloides*
*Mucor circinelloides*	*Rhizomucor regularior, Rhizomucor variabilis var. regularior*
*Mucor griseocyanus*	*Mucor circinelloides f. griseocyanus*
*Mucor irregularis*	*Rhizomucor variabilis*
*Mucor janssenii*	*Mucor circinelloides f. janssenii*
*Mucor lusitanicus*	*Mucor circinelloides f. lusitanicus*
*Rhizopus arrhizus* (incl. var. *delemar*)	*Rhizopus oryzae*
*Rhizopus microsporus*	*Rhizopus microsporus* var. *azygosporus*, var. *chinensis*, var. *oligosporus*, var. *rhizopodiformis*, var. *tuberosus*

**Table 2 jof-06-00265-t002:** Risk factors for mucormycosis. In several studies there are multiple risk factors so the total is more than 100%.

Reference	Characteristics of Studies					Risk Factors/Underlying Diseases (%)							
	Countries of Origin of Cases	Prospective Study	Multicenter Study	Time Period	Total no. of pts	DM	HM	HSCT	SOM/ SOT	AI/CO	Trauma ^#^	HIV	None
Roden et al. 2005 [8]	Global	No	Yes	1940–2003	929	36	15.8	5	1/7	1	8 ^#^	2	19
Jeong et al. 2019 [9]	Global	No	Yes	2000–2017	851	40	32		1/14	3/33	20		18
Skiada et al. 2011 [14]	Europe	Yes	Yes	2005–2007	230	17	44		5/4	44	17	2	8
Lanternier et al. 2012 [10]	France	No	Yes	2005–2007	101	23	50	12	2/3	13	18	1	1
Pagano et al. 2009 [27]	Italy	Yes	Yes	2004–2007	60	18	62	3	8/	3/50	2	17	3
Kontoyiannis et al. 2016 [18]	USA	No	Yes	2005–2014	555	52	40	11	6/15	NA	4	2	NA
Nucci et al. 2019 [28]	South America	No	Yes	1960–2018	143	42	11	2	/13	NA	20	2	7.7
Corzo-Leon et al. 2017 [12]	Mexico	No	Yes	1982–2016	418	72	17		1/	1	2.3	0.7	4
Chakrabarti et al. 2006 [5]	India	No	No	2000–2004	178	73.6	1.1		/0.6	1.7	7.3	0.6	11.8
Chakrabarti et al. 2009 [20]	India	Yes	No	2006–2007	75	44	9		/5	29	11	1	3
Prakash et al. 2019 [29]	India	Yes	Yes	2013–2015	303	56.8	6		/6	9.9	10	-	10.5
Patel et al. 2020 [11]	India	Yes	Yes	2016–2017	465	74	8	1	1.5/6.5	/3.7	6.9	-	11.8
Dolatabadi et al. 2018 [30]	Iran	No	Yes	2008–2014	208	75	3	2	3/3	NA	4	-	2
Vaezi et al. 2016 [31]	Iran	No	Yes	1990–2015	98	48	6	1	/23	NA	1	-	10
El Zein et al. 2018 [32]	Lebanon	No	No	2008–2018	20	35	65		/5	70	-	-	-
Kennedy et al. 2016 [33]	Australia	No	Yes	2004–2012	74	27	48.6	18	3/11	12/ 53	23		11
Stemler et al. 2020 [34]	Middle East and North Africa	No	Yes	1968–2019	310	49.7	16.5		2/17	21.6	12	0.3	5.8

# Penetrating trauma and surgery. SOM = Solid organ malignancy, SOT = Solid organ transplantation, AI = Autoimmune disease, CO = corticosteroids.

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
