# Peer review of "Epidemiology and Diagnosis of Mucormycosis: An Update"

_jof, 2020, doi:10.3390/jof6040265_

Round 1

Reviewer 1 Report

The manuscript titled “Epidemiology and Diagnosis of mucormycosis: an update” is a welcome work that can help to summarize the extensive knowledge rapidly accumulated in the last few years. The number of previous papers that have been reviewed and the quality of the presentation in this revision are impressive. Therefore, I directly recommend publication. I have only one minor suggestion. Mucormycosis is currently described with three major features: 1, it is a highly deadly disease; 2, it is extremely resistant to our current antifungal compounds; 3, and it is considered as an emerging disease. The two first features have been adequately addressed in this work. However, I miss more information about the emergence of mucormycosis. In the past, it was considered a rare infection, but the unquestionable increasing number of cases makes mucormycosis a current emerging disease. The question is: are there more cases because of the emerging of new pathogenic species, or do we have more cases because the diagnosis has greatly improved? It will be appreciated if a summary of the available information in the related literature is included in this review. Also, the author could argue their own opinion in the discussion section. Last, I suggest a few corrections of typos and grammar mistakes for final polishing.

109 an hematological ==> a hematological

316 related with==> related to

330 nonpigmented==>non-pigmented

339 applied on==> applied to

393 gene rnl==> gene rnl

429 these finding==> these findings

488 applied on==> applied to

Author Response

Dear reviewer,

Thank you for reviewing our manuscript. These are the changes we made, according to your comments:

- lines 80-89: in order to show that mucormycosis is an emerging disease, we wrote:

All the above publications highlight the fact that mucormycosis is an emerging disease. Roden et al. included a figure where the rise of incidence over 6 decades (1940-1999) is clearly shown (8). The same authors reported the rise of percentages of cases documented by culture, by decade. The percentage was 50% in the 1980s, and 71% in the 2000s. This may indicate that, in addition to an actual increase of the incidence of cases, better diagnostic methods or increased awareness of the disease play an important role. The rising trend of mucormycosis was also shown by Chakrabarti et al. who published three consecutive studies from a single center in India; the incidence of mucormycosis was 12.9 cases/year over the first decade [19], 35.6 cases/year over a 5-year period [5] and, during an eighteen months period, 50 cases/year [20].

  • line 109 is now line 143: "an hematological" was changed to "a hematological"
  • line 316 is now line 459: "related with" was changed to "related to"
  • line 330 is now line 473: "non pigmented" was changed to "non-pigmented"
  • lines 339 and 488 are now lines 494 and 719: "applied on" was changed to "applied to"
  • line 393 is now line 566: "gene rnl" was changed to "gene rnl"
  • line 429 is now line 602: "these finding" was changed to "these findings"

Reviewer 2 Report

The review by Skiada et al. is well-written and well-organized.

General comments:

none

Specific comments:

I noticed duplication among references 4, 38, and 39.  Please review and correct all references.

Perhaps the addition of a table which includes all the current Mucorales species, with their previous taxonomic names, would add to this paper.

Author Response

Dear reviewer,

Thank you for reviewing our manuscript and sending us these useful suggestions. We corrected the references which were in triplicate:

  • Reference 4, was deleted, and a paper by Slavin et al. was inserted.
  • References 38 and 39, are now references 41 and 42. Reference 41 is the one by Park et al. and reference 42 is now by Pappas et al., also about TRANSNET.

lines 91-94: After your suggestion about adding the species, we wrote:

"The order Mucorales comprises 261 species in 55 genera, 38 of which have been associated with human infections [21]. Their taxonomy has changed widely the last years due to molecular phylogenetic studies and, inevitably, some taxa have undergone several name changes as well. The current nomenclature of these taxa is presented in Table 1." On the inserted table one can see the species which now have new names. We have not included all the species which produce mucormycosis in humans, because these are mentioned extensively in the text.

The recent reference by Walther et al. has been added (reference 21).